# *MsYSL6*, A Metal Transporter Gene of Alfalfa, Increases Iron Accumulation and Benefits Cadmium Resistance

**DOI:** 10.3390/plants12193485

**Published:** 2023-10-05

**Authors:** Miao Zhang, Meng-Han Chang, Hong Li, Yong-Jun Shu, Yan Bai, Jing-Yun Gao, Jing-Xuan Zhu, Xiao-Yu Dong, Dong-Lin Guo, Chang-Hong Guo

**Affiliations:** Heilongjiang Provincial Key Laboratory of Molecular Cell Genetics and Genetic Breeding, College of Life Science and Technology, Harbin Normal University, Harbin 150025, China; zhangmiaokid669@163.com (M.Z.); cmh6650@163.com (M.-H.C.); lihong1232580@163.com (H.L.); syjun2003@126.com (Y.-J.S.); baiyan789@163.com (Y.B.); jingyun20000528@163.com (J.-Y.G.); zhujingxuan99@163.com (J.-X.Z.); dx20211006@163.com (X.-Y.D.)

**Keywords:** cadmium stress, transporter, gene expression, Fe translocation, antioxidant activity

## Abstract

Iron (Fe) is necessary for plant growth and development. The mechanism of uptake and translocation in Cadmium (Cd) is similar to iron, which shares iron transporters. Yellow stripe-like transporter (YSL) plays a pivotal role in transporting iron and other metal ions in plants. In this study, *MsYSL6* and its promoter were cloned from leguminous forage alfalfa. The transient expression of MsYSL6-GFP indicated that MsYSL6 was localized to the plasma membrane and cytoplasm. The expression of *MsYSL6* was induced in alfalfa by iron deficiency and Cd stress, which was further proved by GUS activity driven by the *MsYSL6* promoter. To further identify the function of *MsYSL6*, it was heterologously overexpressed in tobacco. *MsYSL6*-overexpressed tobacco showed better growth and less oxidative damage than WT under Cd stress. *MsYSL6* overexpression elevated Fe and Cd contents and induced a relatively high Fe translocation rate in tobacco under Cd stress. The results suggest that *MsYSL6* might have a dual function in the absorption of Fe and Cd, playing a role in the competitive absorption between Fe and Cd. *MsYSL6* might be a regulatory factor in plants to counter Cd stress. This study provides a novel gene for application in heavy metal enrichment or phytoremediation and new insights into plant tolerance to toxic metals.

## 1. Introduction

Iron (Fe) is an essential micronutrient playing an active role in plant photosynthesis, antioxidant defense systems, electron respiration, and embryogenesis. Under iron deficiency, plants show typical chlorosis symptoms and hampered crop yield and productivity [1]. Plants develop efficient absorption strategies to support the requirement of sufficient iron. Iron abundance in plants is tightly regulated by iron uptake, translocation, and recycling. In addition to playing an important role in the uptake process, iron transporters are responsible for the transport and distribution of iron in plants [2]. Cadmium (Cd) is one of the most phytotoxic elements that negatively affects plant metabolism, growth, and development and indirectly contributes to ROS formation by altering the antioxidant system in plants, reducing crop failure [3,4,5,6]. Since there is no Cd-specific transporter in plants, Cd is supposed to be transported in plants via cation transport systems. Being closely similar to the essential metal element Fe, Cd is often taken up by Fe transporters. For example, NRAMPs in *Arabidopsis thaliana*, *Oryza sativa* [7,8,9], and *Sedum alfredii* [10], IRTs in *Oryza sativa* [11], and OPT and ZIP in *Arabidopsis thaliana* [12,13] transport both Fe and Cd. Cd distribution throughout a plant is an intricate process controlled by root uptake, root-to-shoot translocation through the xylem, and the redistribution of Cd from the leaves to sink tissues [4,14]. The contribution of cation transporters to the management of stress is beneficial in plant improvement programs [15]. Metal transporter genes can be potentially used for engineering genotypes for phytoremediation or minimizing Cd in crops. The overexpression of *AtHMA3* [16], *OsNramp1* [8], *NtPIC1* [17], and *Miscanthus sacchariflorus MsYSL1* [18,19] enhances plant tolerance to Cd. The sodium/calcium exchanger-like gene *TaNCL2-A* plays a role in alleviating the toxic effects of Cd in conjunction with salinity and osmotic stress in Arabidopsis [20].

Cd competes with essential macro- and microelements at their absorption sites, disrupting the homeostasis of crucial microelements [6,21]. It has been postulated that Cd toxicity can be attributed, at least in part, to the perturbation of the metabolism of Fe [22]. The chlorosis of plant leaves under Cd stress is similar to the typical symptoms of iron deficiency [23]. Cd induces moderate-to-strong iron deficiency in leaves, which strongly affects photosynthesis [24]. Previous studies revealed that Cd significantly decreased the Fe content in shoots, but increased the Fe content in roots [22,25]. Fe significantly accumulated in root apoplasts and root cell walls under Cd stress [10]; Fe played a significant role in alleviating the damage caused by Cd toxicity through reducing Cd accumulation and increasing Cd detention, photosynthesis protection, antioxidant defense capacity enhancement, electrolyte leakage amelioration, and other ion status regulation [26,27,28,29]. Limiting Fe uptake through the downregulation of Fe acquisition mechanisms confers a Si-mediated alleviation of Cd toxicity [30]. The Fe regulation transcript factors (*FIT*s) *AtbHLH38*, *AtbHLH39*, *bHLH104*, and *BTS* are involved in Cd tolerance [22,31,32]. Although the Cd-induced increase in root Fe contents has been described, and several possible explanations for such observations including impaired Fe uptake and translocation have been proposed, the underlying mechanism of Cd-induced changes in Fe homeostasis within plants is still not fully understood.

The yellow stripe-like transporter (YSL) family has attracted attention in recent years for its function in iron uptake from soil or iron translocation throughout whole plants [19]. During the last two decades, YSLs have been revealed to transport iron complexes with specific Fe chelators and phytosiderophores. YSLs participate in Fe homeostasis in graminaceous and nongraminaceous plants with distinguished functions in each plant type. In graminaceous plants, most of the YSLs take up the Fe^3+^-mugineic acid (MA) complex [33,34]. Some YSLs transport Fe^2+^-Niacinamide (NA) or Fe^3+^-2′-deoxymugineic acid (DMA) over long distances and enable the distribution of Fe within the plant leaves, phloem, and reproductive organs [35,36,37]. In nongraminaceous plants, YSLs transport the Fe–NA complex over long distances and intracellularly distribute Fe in plants [38,39,40]. Beyond the essential functions in Fe transport, evidence supports the notion that YSLs are mediated in the processes of transporting Cu, Mn, Ni, and Cd in plants [19,41,42,43]. *YSL1* and *YSL3* in *Arabidopsis thaliana* [40], *SnYSL3* in *Solanum nigrum* [44], *BjYSL7* in *Brassica junce* [45], *YSL3* in peanut [46], *MsYSL1* in *Miscanthus sacchariflorus* [18], and *ZmYS1* in maize [47] were reported to be involved in the uptake, distribution, and translocation of Cd.

Leguminous plants can grow in poor and degraded soils. Meanwhile, perennial grasses occupy diverse soils worldwide, including many sites contaminated with heavy metals [26], whereas studies on Fe and Cd interaction have minimal information reported in legumes [48]. Alfalfa (*Medicago sativa* L.) is an essential forage of biological nitrogen fixation, biofuel, and animal feed. Investigators have supported the idea that alfalfa reaches a new steady state to acclimate under chronic Cd stress by adequately adjusting its metabolic composition [30,49]. An increased abundance of xylogalacturonan of *Medicago sativa* after long-term exposure to Cd might hinder Cd binding in the cell wall and is an important factor during tolerance acquisition [50]. At the same time, reducing Cd bioaccumulation and controlling Cd distribution in alfalfa deserve more attention. The overlap of Fe and Cd transport in previous reports reflects the importance of the transporter’s function. The overexpression or elimination of transporter reduces Cd uptake in plants [51].

Moreover, several *YSL* genes have been found to enhance Cd resistance ability in transgenic plants [18,45]. It is beneficial to explore the potential of *YSL* in metal transport for genetic engineering. Several *YSL* genes can assist plants in participating in the process of Cd uptake, transport, and accumulation [52]. There are also some YSLs that can help the zinc–iron biofortification of wheat [53]. It was found that the rice YSL family is closely related to the evolution and function of other plants [54]. Whether YSL is involved in Cd uptake and transport has not been investigated in alfalfa yet. Therefore, we isolated iron transport genes *MsYSL6* from alfalfa, focusing on the Fe and Cd contents and the Cd-tolerant ability in *MsYSL6* transgenic plants. This research will provide new insight into plant metal transporters in heavy metal enrichment or phytoremediation.

## 2. Results

### 2.1. MsYSL6 Belongs to the YSL Transporter Family

The open reading frame (ORF) of *MsYSL6* (LOC_MG673951.1) was cloned from alfalfa by PCR (Appendix A). The full length of *MsYSL6* ORF contains 2 028 bp, encoding 675 amino acids. MsYSL6 protein harbors 12 transmembrane domains (Appendix A). The phylogenetic tree was constructed using data of MsYSL6 and other 23 YSL proteins from eight plant species (Figure 1). MsYSL6 belonged to a branch with AtYSL6, OsYSL6, MtYSL6, GmYSL6, GsYSL6, VaYSL6, CcYSL6, and CaYSL6, indicating a close relationship in these YSLs. These findings indicated that MsYSL6 belonged to the YSL transporter family.

### 2.2. The Subcellular Localization of MsYSL6

The subcellular localization of MsYSL6 was investigated using a fusion protein with GFP (Figure 2). The pBWA(V)HS-*MsYSL6*-Glosgfp was transformed into tobacco leaf cells. Apparent green fluorescence signals of MsYSL6-GFP were observed in the cell membrane and protoplast. The tobacco leaf cells expressing GFP alone exhibited weak and dispersive fluorescence inside the cells (Figure 3). Overlay images were created and showed that MsYSL6-GFP-mediated fluorescence does not co-localize with the chlorophyll-mediated fluorescence.

### 2.3. The Expression of MsYSL6 in Alfalfa

To gather insight into the expression of the *MsYSL6* gene in alfalfa, the transcript profiles of *MsYSL6* were monitored in alfalfa under iron deficiency or Cd treatment, with nonstress treatment as the control. *MsYSL6* showed tissue-specific expression in alfalfa in the control. The expression level was higher in the leaves than that in the roots and the stems. In alfalfa roots, the expression of *MsYSL6* was significantly upregulated by iron deficiency and Cd stress at 0.05 and 0.01 levels, respectively. In particular, *MsYSL6* expression increased nearly 60 folds in alfalfa roots under Cd treatment. In alfalfa stems, the expression of *MsYSL6* was significantly upregulated by iron deficiency and Cd stress at 0.05 level. Although the expression was higher than that in the roots and the stems, the expression of *MsYSL6* showed no significant differences in leaves under these three conditions (Figure 4A).

### 2.4. The GUS Activity

In order to determine the specific expression of *MsYSL6*, we fused the *MsYSL6* promoter in-frame to β-glucuronidase (GUS) reporter (Appendix A). The pBI121-*MsYSL6*pro::GUS construct was transformed in alfalfa and induced hairy roots (Appendix A). The alfalfa with hairy roots were grown in a medium containing 75 µM CdCl_2_, with no Cd treatment as the control. The alfalfa hairy roots exhibited blue when the medium contained no Cd, and exhibited strong blue when the medium contained Cd. The alfalfa cotyledons exhibited no staining when the medium contained no Cd, and exhibited blue when the medium contained Cd (Figure 4B). The GUS activity driven by *MsYSL6*pro was enhanced by Cd indicating that *MsYSL6* responded to Cd stress, which is consistent with the upregulated expression of *MsYSL6* detected by qRT-PCR.

### 2.5. The Growth of MsYSL6 Transgenic Tobacco

To screen the effect of *MsYSL6* on Cd tolerance, the T_3_ seeds of three *MsYSL6* overexpress lines (*MsYSL6*OE), namely, L5, L8, L9, and WT, were grown in the medium containing 50 µM CdCl_2_, 75 µM CdCl_2_, and 100 µM CdCl_2_, with no Cd as the control. A preliminary observation showed that *MsYSL6*OE germination was similar to WT plants when grown in no Cd medium. The germination of *MsYSL6*OE and WT decreased following the increase of Cd concentration, and the germination of *MsYSL6*OE lines was significantly less affected than that of WT (Appendix A). The WT plant grown on 75 µM CdCl_2_ medium exhibited shorter and smaller roots, while *MsYSL6*OE exhibited a normal root phenotype with minimal stunted growth. The *MsYSL6*OE plants showed significantly higher fresh weight and longer roots than the WT (Figure 5).

### 2.6. The Chlorophyll Content

The chlorophyll content decreased in *MsYSL6*OE and WT plants under Cd treatment more than that of the control. The chlorophyll content of *MsYSL6*OE leaves was significantly higher than that of WT in the control at 0.05 level. The chlorophyll content of L5 and L8 leaves was significantly higher than that of WT under Cd treatment at 0.05 level. The higher chlorophyll content showed that the *MsYSL6*OE transgenic tobacco was more tolerant to Cd stress (Figure 6A).

### 2.7. The MDA Content

The malondialdehyde (MDA) content showed no difference in *MsYSL6*OE tobacco and WT in the control. Cd treatment increased the MDA content in the *MsYSL6*OE tobacco and WT plants. The MDA content in L5, L8, and L9 was significantly lower than that of WT under Cd treatment at 0.05 level (Figure 6B). The results showed that overexpress of *MsYSL6* protects plants from damage and leakage in the membrane caused by Cd stress.

### 2.8. The DAB and NBT Staining

Under Cd treatment, DAB staining showed that the leaf of WT was deeper brown while the leaves of L5, L8, and L9 were shallow brown. The staining was scattered at the leaves of L5, L8, and L9, and the brown color was less than that of WT. NBT staining showed that the leaf of WT was deeper blue, while the leaves of L5, L8, and L9 were shallow blue (Figure 7). The results represent that *MsYSL6*OE tobacco has less ROS production than WT under Cd stress.

### 2.9. The Cd Content and Translocation Rate

The Cd accumulation in *MsYSL6O*E plants under 75 µM Cd treatment with a significant increase was observed compared with the WT plants (Figure 8A). The Cd content in L5, L8, and L9 roots was significantly higher than that of the WT at 0.05 level. The Cd content in shoots of L5 and L8 was significantly higher than that of the WT at 0.05 level, which showed no difference in L9 from that of the WT (Figure 8A). The Cd translocation rate was 31%, 34%, 38%, and 33% in L5, L8, L9, and WT, respectively.

### 2.10. The Fe Content and Translocation Rate

In the roots, the Fe content of *MsYSL6*OE lines was significantly higher than that of WT under Cd or non-Cd treatment at 0.05 level. Notably, the Fe content in roots was increased in all lines by Cd treatment, and the increase of *MsYSL6*OE plants was greater than that of WT (Figure 8B). In the shoots, the Fe content was significantly higher in L8 and significantly lower in L9 than that in WT under non-Cd treatment; the Fe content was significantly higher in L5 and L8 than that in WT under Cd treatment at 0.05 level; the Fe content increased in *MsYSL6*OE shoots and decreased in WT shoots by Cd treatment (Figure 8C). Under non-Cd treatment, the Fe translocation rate accounted for 15%, 24%, and 13% in L5, L8, and L9, respectively, both lower than 29% in WT. Under Cd treatment, the Fe translocation rate accounted for 24%, 29%, and 16% in L5, L8, and L9, respectively, both lower than 34% in WT.

### 2.11. The Cu, Mn, and Zn Content

The Cu, Mn, and Zn content were also screened in *MsYSL6*OE plants and the WT plants. In plant roots, Cu contents were significantly higher and Mn contents were significantly lower in *MsYSL6*OE lines than that of WT plants under no-Cd treatment at 0.05 level; Cu contents and Mn contents in L5 and L8 were significantly higher and Mn content in L9 was significantly lower than that of the WT plants under Cd treatment at 0.05 level; Zn contents of *MsYSL6*OE lines showed no difference with the WT plants under all treatments (Figure 9A–C). In shoots, the Cu and Zn contents in L8 and L9 were significantly higher and Mn content in L5 shoots was significantly lower than that of the WT plants under no-Cd treatment at 0.05 level; the Mn contents in L5, L8, and L9 shoots were significantly lower than that of the WT plants at 0.05 level (Figure 9D–F). In general, Cu, Mn, and Zn contents in the roots and Cu contents in the shoots decreased, while Mn and Zn contents in the shoots of *MsYSL6*OE plants and the WT plants increased by Cd treatment.

## 3. Discussion

Although not essential for plant growth, Cd is readily taken up by roots and accumulated in plant tissues to high levels [26]. Cd uptake and transport in different plant species is linked to Fe due to their similar chemical characteristics. It has been reported that the iron transporter *OsNRAMP5* contributes to the uptake of Cd from the soil, and the total Cd content in the *OsNRAMP5*i plants decreases [9]. The upregulation of *IRT1* facilitated Cd uptake by roots, thus increasing the Cd concentration in plant tissues [32]. Transcript levels of *IRT1* are very low and transcript levels of *HMA2* are strongly elevated in *Arabidopsis halleri* from the most highly heavy metal-contaminated soil, which can account for its altered Cd handling [55]. *ABC*, *NRAMP*, and *ZIP* genes might play important roles in different levels of Cd accumulation in sunflower cultivars [56]. *MsYSL1 SnYSL3* and *BjYSL7* may be essential transporters for diverse metal–NAs to participate in the Cd detoxification by mediating the reallocation of other metal ions [18,40,45]. Studies have shown that most YSLs may function in the intracellular transport, translocation, and distribution of Fe and other metals in *Arabidopsis thaliana* and *Oryza sativa*. In this study, we obtained a novel YSL transporter gene, *MsYSL6,* from alfalfa. MsYSL6 belonged to a branch with AtYSL6, OsYSL6, and several YSL6s in leguminous plants, indicating a close relationship in these YSLs. A fundamental role has been demonstrated for Arabidopsis YSL6 in managing chloroplastic iron [38]. OsYSL6 was suggested to be a Mn–NA transporter [42].

Transporters play specific biological roles in mineral transport within the tissues where they are expressed. Data indicated that YSL expression was detected in three main territories: vascular tissues throughout the plant, pollen grains, and seeds [35,39,57,58]. *MsYSL6* was expressed in all vegetative tissues of alfalfa and was most expressed in the leaf. However, *MsYSL6* was induced by iron deficiency and Cd in alfalfa roots, consistent with *MsYSL1*, *SnYSL3,* and *BjYSL7*, suggesting an important role for *MsYSL6* in response to Cd stress, possibly functioning in Cd uptake [18,43,44]. The altered patterns of *MsYSL6* expression might be a response to Cd stress.

In plants, Cd toxicity disrupts photosynthesis, causes the production of reactive oxygen species (ROS), and increases antioxidant enzyme activity [6,21,27,59,60,61]. The present study also found that *MsYSL6* overexpression reduced ROS accumulation and mitigated the adverse effects on the growth of tobacco under Cd stress, in accordance with previous studies. The specific mechanism of *MsYSL6* conferring tolerance to Cd stress into tobacco might be involved in metal distribution.

The Fe-biofortified lentil genotype exhibited Cd tolerance by inciting an efficient antioxidative response to Cd toxicity [19]. Fe significantly reduced Cd transfer towards rice grains, which might be attributed to a sharp decrease in the proportion of bioavailable Cd in leaves [62]. The significantly high Fe content in *MsYSL6* transgenic tobacco indicated the primary function of *MsYSL6* in Fe uptake from the medium. Increasing supply of Fe remarkably reduced Cd accumulation in rice shoots, mainly because of inhibited translocation of Cd from rice roots to shoots [63]. Fe deficiency increased Cd uptake and accumulation in peanuts but decreased Cd translocation from roots to shoots [64]. In rice, low Fe or excess Fe facilitated the uptake of Cd in rice roots, as low Fe upregulated the expression of Cd-transport-related genes, and excess Fe enhanced Cd enrichment on the root by iron plaque [63]. Thus, the accumulation of Cd in *MsYSL6* transgenic tobacco might be due to the high expression level of *MsYSL6* in this study.

Many studies have found that Cd stress affects Fe concentration in plants. Cd-induced Fe accumulation in roots is mediated by upregulating Fe transporter genes such as *IRT*1 [65]. The Cd and Fe accumulation in plants could be partially due to the crucial ligands for metal chelation [66]. Like *AhOPT3*/*6*/*7* and *AhYSL1*/*3* in peanuts, *MsYSL6* might be involved in the transport of Fe and/or Cd and Fe/Cd interactions [67]. Cd damages the photosynthetic apparatus, as well as the “Fe-deficiency-like” symptoms, and the supply of Fe nutrients saves the photosynthesis symptoms caused by Cd [68,69]. Under Cd stress, high Fe concentrations in *MsYSL6* confer the plant tolerance to Cd toxicity. We infer that *MsYSL6* might have a dual function in the absorption of Fe and, at the same time, Cd. It plays a role as a regulatory factor in the competition absorption between iron and Cd. *MsYSL6* increased the ratio of iron to Cd in the total amount and transferred more Fe to the above-ground part of the plant, thereby reducing the oxidative damage of Cd to the plant and resulting in the improvement of tolerance to Cd toxicity in *MsYSL6* transgenic tobacco lines.

The metal translocation rate represents the metal translocate capacity in plants. The Cd translocation ratio showed little difference in all tested plant lines. From the results, we hypothesize that *MsYSL6* might possibly contribute to the Cd uptake while not likely involved in Cd transport from roots to shoots. In this study, overexpression of *MsYSL6* did not increase the Fe transport rate, suggesting that *MsYSL6* might not be involved in the translocation of Fe from roots to shoots, while Cd stress increased the Fe transport rate in *MsYSL6*OE lines, which is similar to the idea that Cd exposure increases the root-to-shoot translocation ratios of Fe in *MsYSL1* and *SnYSL3* overexpressing *Arabidopsis thaliana*. The relationship between Cd stress and Fe uptake ability deserves further investigation [66]. Additionally, the absorption and transport of Mn, Zn, and Cu in *MsYSL6* transgenic tobacco were affected by Cd stress. Specialized cation transport mechanisms have been developed in plants, which can maintain a balance between the deficiency and toxicity of these ions [15]. The state of these metals could also have an effect on plant growth and is worth further study.

The phytosiderophores chelate iron were suggested to prevent graminaceous plants from Cd toxicity. In maize, phytosiderophore provides an advantage under Cd stress relative to Fe acquisition via ferrous Fe [47]. The nicotinamide synthase (NAS) synthesizes nicotinamide (NA), which plays an essential role together with YSLs in iron homeostasis in nongraminaceous plants. It has also been reported that Cd induces the expression of the *NAS* gene in *Medicago truncatula* [70]. *MsYSL6* may also achieve the effect of alleviating plant Cd toxicity by co-acting with *NAS*, despite reports claiming that YSL6 in *Arabidopsis thaliana* has an essential role in chloroplast development, likely in the transport of Fe required for chloroplast development [36]. In another study, AtYSL6 was localized to vacuole membranes and to internal membranes [71]. In this study, MsYSL6 was found to have no chloroplast localization properties, so it is unlikely that MsYSL6 is involved in Cd detoxification through this pathway. Our results support the idea that MsYSL6 may function as a Cd/Fe transport gene and may be useful for genetic engineering in cultivating Fe-fortified or Cd-tolerant crops.

## 4. Materials and Methods

### 4.1. Plant Growth Conditions

Alfalfa (*Medicago sativa* L. cv. Zhaodong) seeds were soaked in 70% ethanol and 10% NaClO for 10 min, then washed with H_2_O three times. Seeds were placed onto sterile Whatman paper and germinated for 3 d in the dark. After germination, seedlings were transferred into 1/2 Hoagland’s solution. The solution was renewed every 2 d. The seedlings were grown at 25 °C with 16 h light/8 h darkness. The wild-type tobacco (*Nicotiana tabacum*) seeds were grown on a sterile MS medium at 25 °C with 16 h light/8 h darkness. For detection of gene expression, 30-day-old hydroponic alfalfa seedlings were treated with iron deficiency (no Fe supply, -Fe), 0.5 mM CdCl_2_ (Cd), and the control (no stress, Ck) for 4 days. For detection of the tolerance to Cd stress, T_3_ of *MsYSL6*OE and WT tobacco seedlings were germinated and cultivated in MS medium for four weeks, and then transferred onto MS medium containing 75 µM CdCl_2_ (Cd) for 10 d, with no CdCl_2_ MS medium as the control (Ck).

### 4.2. RNA and DNA Extraction

The total RNA of plant tissues was isolated using the RNA prep Pure Plant Kit 175 (TianGen Biotech, Beijing, China). RNA derived from the leaf was reverse transcribed into cDNA by using Easy Script One-Step DNA Removal and cDNA Synthesis SuperMix (TransGen 177 Biotech, Beijing, China). The total RNA quality was performed by visualization on a 1 × TBE gel stained with ethidium bromide. Total DNA was extracted using a Plant Kit (OMEGA, New York, NY, USA, D3485-01).

### 4.3. Cloning of MsYSL6 and MsYSL6pro

Alfalfa cDNA was used as the template to amplify the open reading frames (ORFs) of the *MsYSL6* gene. The *MsYSL6* gene amplified primers were designed according to alfalfa iron deficiency transcriptome data (Appendix A). The PCR system was as follows: template 1 µL, each primer 0.4 µL, ExTaq DNA polymerase (5 U/µL) 0.05 µL, dNTP mix (2.5 mM) 0.8 µL, 10 × PCR Buffer 1 µL, add ddH_2_O to 10 µL. PCR was performed using the following program: 94 °C for 5 min, 94 °C for 30 s, 55 °C for 30 s, 72 °C for 2.5 min, 95 °C for 15 s, for 30 cycles; 16 °C for 1 h. The PCR products were purified with a Gel DNA Purification Kit (OMGAE), and then cloned into pMD18T plasmids for sequencing. The amino acid sequence of MsYSL6 was identified using the SMART website (http://smart.embl-heidelberg.de/ accessed on 11 June 2022). To elucidate the evolutionary relationship, a BLASTP search was performed in the NCBI nr (nonredundant protein sequences) database. Amino acid sequences were aligned with MEGA7.0 software. An unrooted neighbor-joining tree was constructed. The transmembrane domains were predicted using the TMHMM Server v.2.0 website (http://www.cbs.dtu.dk/services/TMHMM/ accessed on 11 June 2022). Conserve protein motifs were predicted through the online program MEME (http://memesuite.org/tools/meme accessed on 11 June 2022).

The amplification primers of the *MsYSL6* promoter (*MsYSL6*pro) fragment were designed according to alfalfa genomic data (Appendix A). The PCR system was as follows: template DNA 1 µL, each primer 0.4 µL, ExTaq DNA polymerase (5 U/µL) 0.05 µL, dNTP mix (2.5 mM) 0.8 µL, 10 × PCR Buffer 1 µL, add ddH_2_O to 10 µL. The PCR was performed as follows: 94 °C for 5 min; 30 cycles of 94 °C for 30 s, 58.1 °C for 30 s, and 72 °C for 2 min; and a final extension at 72 °C for 10 min. The PCR products were separated and purified with a Gel DNA Purification Kit (OMEGA). Purified products were then cloned into pMD18T plasmids for sequencing. The cis-elements in *MsYSL6*pro were predicted online by PlantCARE.

### 4.4. Subcellular Localization of MsYSL6

The *MsYSL6* ORF without stop codes was cloned into the pBWA(V)HS-GLosgfp vector to construct a fusion plasmid. Then, the fusion plasmid was introduced into *Agrobacterium tumefaciens* GV3101 by freeze–thaw method and injected into tobacco leaf through the epidermis with a syringe. After cultivation for 2 d, fluorescence signals were recorded using a confocal laser scanning microscope (Nikon C2-ER Laser Scanning Confocal Microscope (Tokyo, Japan)). The green fluorescent protein was excited at 488 nm, and emissions were collected at 510 nm. The chlorophyll autofluorescence was excited at 640 nm, emissions were collected at 675 nm, and overlay images were created.

### 4.5. qRT-PCR

The qRT-PCR was performed by a Real-Time PCR System (Applied Biosystems 7300 Real-Time PCR System, Foster City, CA, USA), using specific primers (Appendix A). *MsActin* and *NtGAPDH* were used as internal controls. The qRT-PCR system was as follows: cDNA 2 µL, primer Mix 1 µL, qPCR Mix 10 µL, 50 × ROX 0.4 µL, ddH_2_O 6.6 µL, total 20 µL. The qRT-PCR was performed as follows: 95 °C for 10 min, 95 °C for 15 s, 55 °C for 30 s, 72 °C for 30 s, 95 °C for 15 s, 60 °C for 30 s, for 40 cycles; 95 °C for 15 s. The qRT-PCR results were quantified using the comparative 2^−ΔΔCt^ method. All of the qRT-PCR experiments were performed with three independent RNA samples.

### 4.6. GUS Assay

The *MsYSL6*pro was cloned to create a C-terminal translational fusion to the reporter gene GUS. The pBI121-*MsYSL6*pro::GUS was introduced into the *Agrobacterium rhizogenes* K599 and then infected on alfalfa roots to induce hairy roots. *MsYSL6*pro::GUS transformed explants were cultured with MS_1_ medium with 50 µM kanamycin for 4 d and then cultured with MS_3_ medium. After hairy root emergence, the transformants were transferred to an MS_3_ medium supplemented with 75 µM CdCl_2_ (Cd) and no stress (Ck) for 15 d. The alfalfa hairy roots were stained for the exam the GUS activity.

### 4.7. Plant Transformation and Selection

*MsYSL6* was digested with XbaI and SmaI and cloned into a plant expression vector pBI121. The *Agrobacterium tumefaciens* GV3101 harboring the pBI121-35S:: *MsYSL6* was transformed into tobacco by leaf disc method. Subsequently, the transgenic T_0_ plants were consecutively self-crossed and detected by PCR and qRT-PCR analysis with *MsYSL6* gene-specific primers. T_3_ seeds were used for further analyses.

### 4.8. Phenotypic and Physiological Analysis

The 2-week-old tobacco seedlings of three *MsYSL6*OE tobacco lines (L5, L8, and L9) and WT were treated with 75 µM CdCl_2_ (Cd) and no stress (Ck) for 15 d. The germinated seeds were recorded and used to calculate the germination rate. The fresh weight and root length of seedlings were measured and photographed. The chlorophyll was extracted by acetone according to Lichtenthaler [72] et al., and the absorption value was measured at 649 nm and 665 nm by an ultraviolet–visible spectrophotometer (Metash Instruments CO. UV-5100). The MDA content was measured at 450 nm according to Reshmi et al.[73].

### 4.9. Oxidative Analysis

The leaves from tobacco were infiltrated with NBT and DAB dyeing solution following published procedures. The leaf was infiltrated with 1 mg/mL nitro blue tetrazolium (NBT) (pH 7.8) for 40 to 60 min in the dark to detect O_2_^-^ accumulation. The leaf was infiltrated with 1 mg/mL diaminobenzidine (DAB) (pH 7.0) for 8 h in the dark to detect H_2_O_2_ accumulation; the leaf was then decolorized by boiling in ethanol: glycerin (3:1) solution for 10 min, photographed after cooling.

### 4.10. Metal Content Determination

The 30-day-old tobacco seedlings of *MsYSL6* transgenic tobacco (*MsYSL6*OE) lines and wild-type plants (WT) were treated with 75 µM CdCl_2_ for 10 d [74]. The shoots and roots of *MsYSL6*OE and WT were collected, and all samples were dried to measure weight. The dried and ground samples were placed into the digestive tube with 4 mL 65% HNO_3_ and heated at 120 °C for 90 min. After cooling, 1 mL 30% H_2_O_2_ was added and it was heated at 120 °C for 90 min. After cooling, 750 µL 30% H_2_O_2_ was added and it was left to stand for 4 h, constant to 50 mL with ddH_2_O. All standard solutions were analyzed with the ICP-OES method using ICP-OES8000 (Perkin Elmer, Waltham, MA, USA). All of the measurements were performed with three independent samples.

### 4.11. Statistical Analysis

All of the experiments were repeated three times independently. Data were analyzed by one-way analysis of variance (ANOVA) with LSD Test (least significant difference). The significance of the data was annotated with asterisks (*) based on the following criteria: *, *p* < 0.05, **, *p* < 0.01.

## 5. Conclusions

Understanding how Fe/Cd accumulates in plants provides guidance for crop cultivation. For the first time, the role in Fe/Cd transport of *MsYSL6* was revealed. This study was a preliminary characterization of the *MsYSL6* gene. Studies are needed to determine the subcellular localization of MsYSL6 protein using markers specific for chloroplasts and various organelles. Although studies showing that Fe/Cd uptake and accumulation mechanisms in plants are similar in some respects, increasing numbers of researchers have found that these mechanisms are distinct between Fe/Cd uptake and accumulation. *MsYSL6* might cooperate with NAS to exercise their biological functions, but it remains unclear. Further study is needed to further clarify these mechanisms.

## Figures and Tables

**Figure 1 plants-12-03485-f001:**
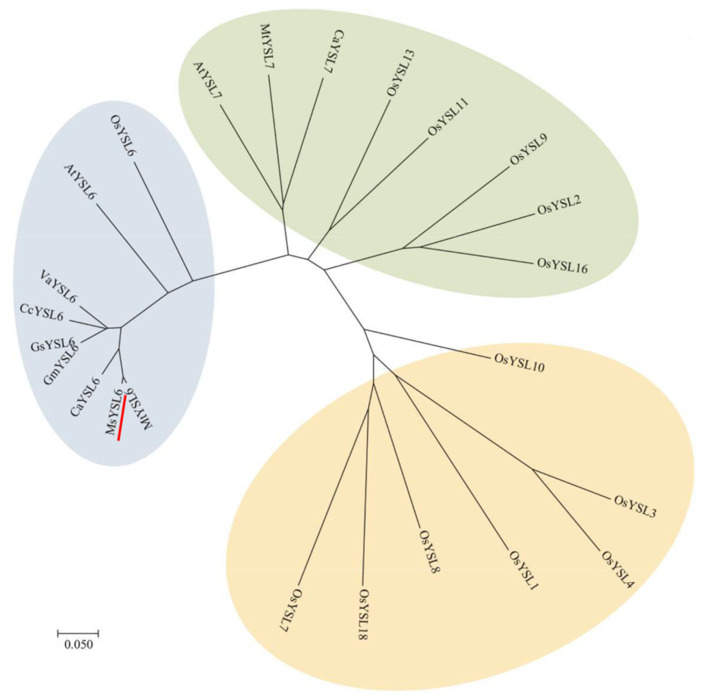
The phylogenetic analysis and domain characteristics of MsYSL6. The phylogenetic tree was constructed using data of MsYSL6 and other 23 YSL proteins from nine plant species, *OsYSL6* (NP_001406189.1), *AtYSL6* (NP_566806.1), *CcYSL6* (LOC109787669), *VaYSL6* (LOC108329245), *GsYSL6* (LOC114398115), *GmYSL6* (LOC100799897), *CaYSL6* (LOC101506832), *MtYSL6* (LOC11438343), *AtYSL7* (NC_003070.9), *MtYSL7* (LOC11415075), *CaYSL7* (LOC11415075), *OsYSL13* (LOC4336441), *OsYSL11* (LOC4336445), *OsYSL9* (LOC4336545), *OsYSL2* (LOC4330161), *OsYSL16* (LOC4336546), *OsYSL10* (LOC4337382), *OsYSL3* (LOC4338223), *OsYSL4* (LOC4338224), *OsYSL1* (LOC4326360), *OsYSL8* (LOC4328078), *OsYSL18* (LOC4327424), and *OsYSL7* (LOC4328077), using methods of neighbor-joining by MEGA 7.0. Red underline highlights the MsYSL6 protein. Ms (*Medicago sativa*); Mt (*Medicago truncatula*); Ca (*Cicer arietinum*); At (*Arabidopsis thaliana*); Os (*Oryza sativa*); Gm (*Glycine max*); Va (*Vigna angularis*); Gs (*Glycine soja*); Cc (*Cajanus cajan*).

**Figure 2 plants-12-03485-f002:**
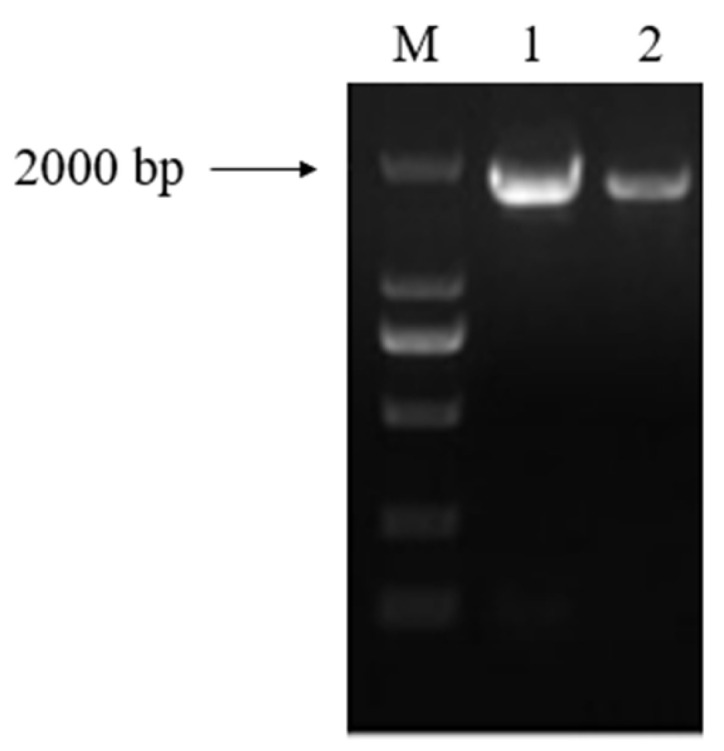
PCR amplification of *MsYSL6* promoter fragment. M, Marker DL 2000; 1–2, *MsYSL6* promoter.

**Figure 3 plants-12-03485-f003:**
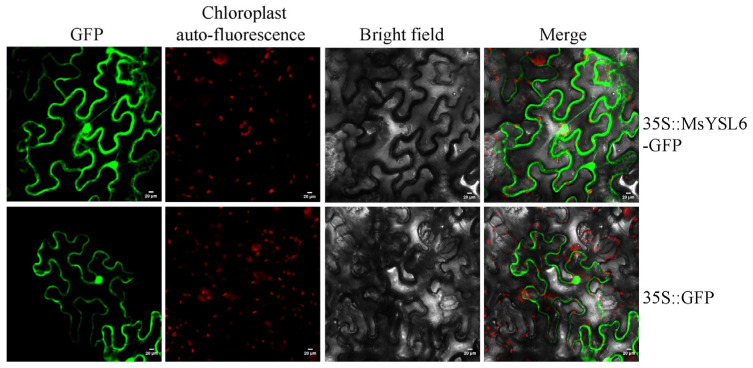
The subcellular localization of MsYSL6. The subcellular localization of MsYSL6 was investigated using a pBWA(V)HS-*MsYSL6*-Glosgfp vector. MsYSL6-GFP fusion protein transiently expressed in tobacco. Left to right: green fluorescence of GFP, red fluorescence of chloroplast spontaneous fluorescence, bright field, and merged microscope images (scale bars: 20 µm).

**Figure 4 plants-12-03485-f004:**
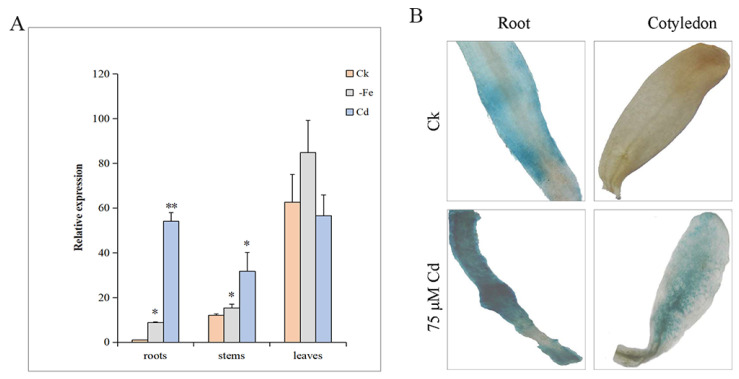
The expression of *MsYSL6* in response to iron deficiency and Cd. The 30-day-old hydroponic young alfalfa seedlings were treated with no Fe supply (–Fe), 0.5 mM CdCl_2_ (Cd), and nonstress as the control (Ck) for 4 days. (**A**) The relative expression of *MsYSL6* in the roots, the stems, and the leaves of alfalfa. Data are means ± s.d. of three experimental replications. Statistical comparison was performed by ANOVA followed by an LSD test. Asterisks indicate significant differences between the treatment and the control (* *p* < 0.05; ** *p* < 0.01). (**B**) The GUS activity induced by *MsYSL6*pro in alfalfa hairy roots. The alfalfa hairy roots were treated with 75 µM CdCl_2_ (Cd) and no stress as the control (Ck) for 15 days.

**Figure 5 plants-12-03485-f005:**
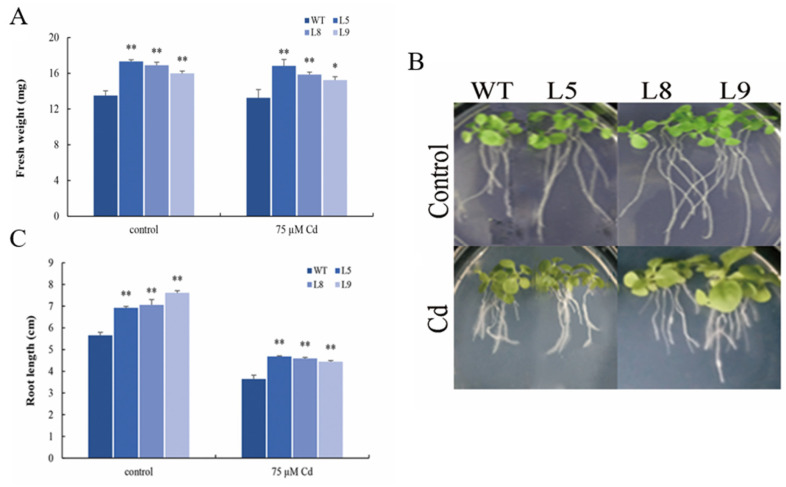
The growth of *MsYSL6*OE tobacco under Cd stress. The 4-week-old tobacco seedlings of three *MsYSL6OE* tobacco lines (L5, L8, and L9) and WT plants were treated with 75 μM CdCl_2_ (Cd) and no stress as the control (Ck) for 10 d. The fresh weight and root length were detected. (**A**) The fresh weight of in *MsYSL6OE* tobacco lines and WT. (**B**,**C**) The root length of in *MsYSL6OE* tobacco lines and WT. The data presented are the means ± s.d. of three experimental replications. Statistical comparison was performed by ANOVA followed by an LSD test. Asterisks indicate significant differences (* *p* < 0.05; ** *p* < 0.01).

**Figure 6 plants-12-03485-f006:**
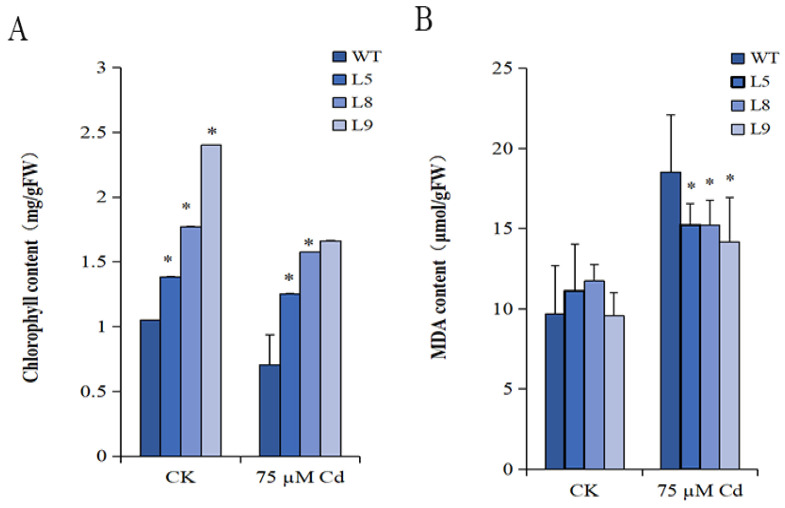
The chlorophyll and MDA content of *MsYSL6*OE tobacco under Cd stress. The 4-week-old seedlings of three *MsYSL6*OE tobacco lines (L5, L8, and L9) and WT plants were treated with 75 µM CdCl_2_ (Cd) and no stress as the control (Ck) for 10 d. (**A**) The chlorophyll content in *MsYSL6OE* tobacco lines and WT. (**B**) The MDA content in *MsYSL6OE* tobacco lines and WT. The data presented are the means ± s.d. of three experimental replications. Statistical comparison was performed by ANOVA followed by an LSD test. Asterisks indicate significant differences (* *p* < 0.05).

**Figure 7 plants-12-03485-f007:**
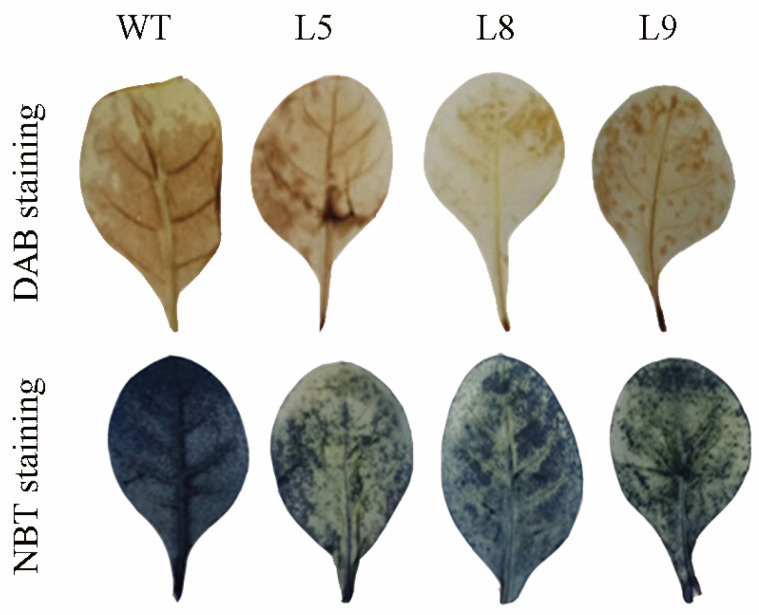
The DAB and NBT staining of *MsYSL6*OE tobacco under Cd stress. The 4-week-old tobacco seedlings of *MsYSL6OE* tobacco lines (L5, L8, and L9) and WT plants were treated with 75 μM CdCl_2_ (Cd) and no stress as the control (Ck) for 10 d. The leaf was stained with nitro blue tetrazolium chloride (NBT) and diaminobenzidine (DAB). The leaf of WT showed deeper staining. The results represent that *MsYSL6OE* tobacco has less ROS than WT under Cd stress.

**Figure 8 plants-12-03485-f008:**
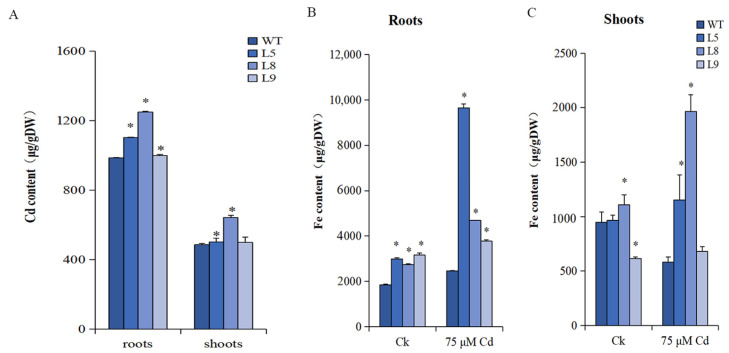
The Cd and Fe contents in *MsYSL6OE* tobacco and WT. The 4-week-old tobacco seedlings of three *MsYSL6OE* tobacco lines (L5, L8, and L9) and WT plants were treated with 75 μM CdCl_2_ (Cd) and no Cd treatment as the control (Ck) for 10 d. (**A**) The Cd content in the roots and shoots of *MsYSL6OE* tobacco lines and WT. (**B**) The Fe content in the roots of *MsYSL6OE* tobacco lines and WT. (**C**) The Fe content in the shoots of *MsYSL6OE* tobacco lines and WT. Data are means ± s.d. of three experimental replications. Statistical comparison was performed by ANOVA followed by an LSD test. Asterisks indicate significant differences (* *p* < 0.05).

**Figure 9 plants-12-03485-f009:**
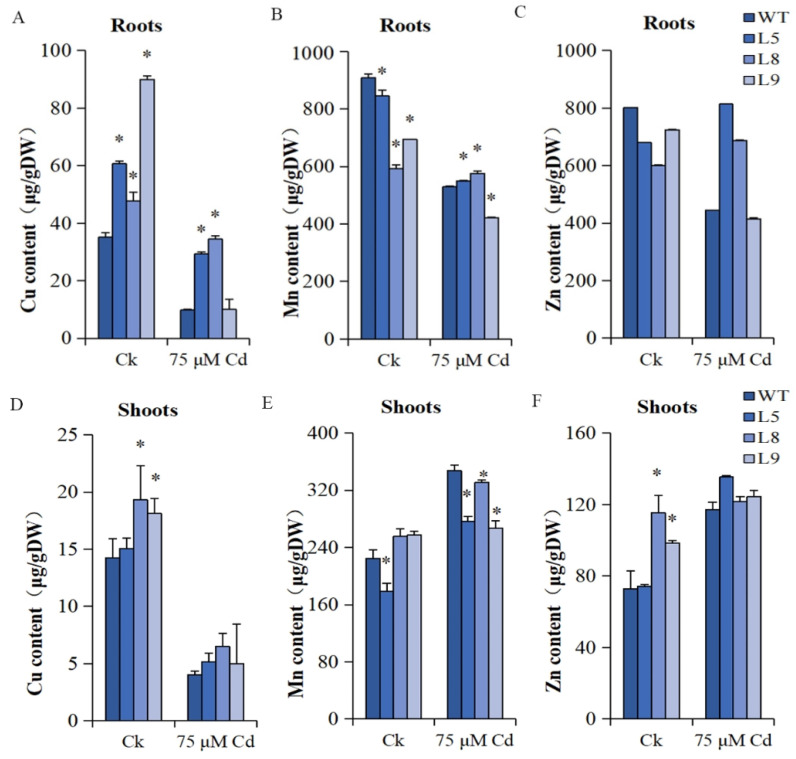
The Cu, Mn, and Zn contents in *MsYSL6*OE tobacco and WT. The 4-week-old tobacco seedlings of three *MsYSL6OE* tobacco lines (L5, L8, and L9) and WT plants were treated with 75 μM CdCl2 (Cd) and no stress as the control (Ck) for 10 d. (**A**) The Cu content in the roots of *MsYSL6OE* tobacco lines and WT. (**B**) The Mn content in the roots of *MsYSL6OE* tobacco lines and WT. (**C**) The Zn content in the roots of *MsYSL6OE* tobacco lines and WT. (**D**) The Cu content in the shoots of *MsYSL6OE* tobacco lines and WT. (**E**) The Mn content in the shoots of *MsYSL6OE* tobacco lines and WT. (**F**) The Zn content in the shoots of *MsYSL6OE* tobacco lines and WT. The Cu, Mn, and Zn contents in the roots and shoots were assessed by ICP-OES. Data are means ± s.d. of three experimental replications. Statistical comparison was performed by ANOVA followed by an LSD test. Asterisks indicate significant differences (* *p* < 0.05).

## Data Availability

Not applicable.

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
