# Peer review of "MsYSL6, A Metal Transporter Gene of Alfalfa, Increases Iron Accumulation and Benefits Cadmium Resistance"

_plants, 2023, doi:10.3390/plants12193485_

Round 1
Reviewer 1 Report
The manuscript submitted by Zhang et al. entitled "MsYSL6, a Metal Transporter Gene of Alfalfa, Increases Iron Accumulation and Benefits Cadmium Resistance" presents an interesting experimental activity related to the Cd traslocation gene activity.
Considering the importance of this study for the implication related to the cultivation on Cd polluted soils, in my opinion, deserve to be published in Plants.
The experimental activity was carried out in an appropriate way, usign widely applied methods well described in the M&M section. Introduction well present the state of the art and the aim of the experiment. The results and their discussion are clearly presented and written. Conclusions are solid and based on the results.
Only minor changes are needed to the manuscript before pubblication, principally related to the form. My specific comments are enclosed in the attached pdf file.

Reviewer 2 Report
In the Ms ‘
MsYSL6, a Metal Transporter Gene of Alfalfa, Increases Iron 2 Accumulation and Benefits Cadmium Resistance” authors showed that MsYSL6 was induced in alfalfa by iron deficiency and Cd stress, which was further 19 proved by GUS activity driven by the MsYSL6 promoter. To further identify the function of MsYSL6, 20 it was heterologously overexpressed in tobacco. MsYSL6 overexpressed tobacco showed better 21 growth and less oxidative damage than WT under Cd stress.
The study is interesting but it has been written very casually, therefore needs major revision before publication.
1. What is Figure 0A?
2. Figure quality is very poor that should be improved.
3. 2.1, to which YSL it was compared, not clearly mentioned in the result.
4. Fig 1 B, I could not see any explanation for this in the result section. Moreover, I think there is no need of this.
5. Why supplementary figures are displayed in the main Ms, for eq. Figure S1A
6. Localization should be studies using markers.
7. Besides, authors failed to discuss the updated reports. I could see numerous recent publications on Cd tolerance. Authors may refer to the Elsevier book ‘Cation transporters in plants’, Chapter 1, Cation transporters in plants: an overview. A recent paper is published in Chemosphere where TaNCL is shown to provide Cd tolerance. These reports need to be discussed in comparision to the current results.
8. Discussion should be focussed on the major findings, and should not be the repeat of results.
9. Similarly, conclusion can not be the repeat of abstract. Author may include gap area and future perspective.
10. Methods also needs significant corrections.
Needs correction.
Round 2
Reviewer 2 Report
Ms is improved. Only minor typos needs to be removed.
Ref 70 seems to be wrongly cited. That may be - Upadhyay, S. K. Cation transporters in plants, Elsevier. 2021.
It may be accepted after minor corrections.
